# Analysis of Therapeutic Decisions for Infantile Hemangiomas: A Prospective Study Comparing the Hemangioma Severity Scale with the Infantile Hemangioma Referral Score

**DOI:** 10.3390/children9121851

**Published:** 2022-11-28

**Authors:** Tong Qiu, Kaiying Yang, Shiyi Dai, Siyuan Chen, Yi Ji

**Affiliations:** 1Division of Oncology, Department of Pediatric Surgery, West China Hospital of Sichuan University, Chengdu 610041, China; 2Pediatric Intensive Care Unit, Department of Critical Care Medicine, West China Hospital of Sichuan University, Chengdu 610041, China

**Keywords:** Hemangioma Severity Scale (HSS), Infantile Hemangioma Referral Score (IHReS), referral, area under the curve (AUC), receiver operating characteristic curve (ROC), Fagan nomogram

## Abstract

Background: In view of the high incidence of infantile hemangioma (IH) in infants and young children, a comprehensive and reasonable evaluation scale for referral is urgently needed. This study compared the influence of the Hemangioma Severity Scale (HSS) and the Infantile Hemangioma Referral Score (IHReS) on treatment decisions for infantile hemangioma patients. Objective: We aimed to establish a reliable and effective evaluation method for referral. Methods: This was a prospective study to determine whether treatment was needed for IH patients after evaluation with the HSS and IHReS. Results: A total of 266 consecutive referred IH patients were evaluated for the risk of IH, and the treatment rate was 80.8%. The area under the curve (AUC) of the subject receiver operating characteristic curve (ROC) of treatment decision making after referral by the HSS was 0.703 (95% CI: 0.634–0.772), and after referral by the IHReS was 0.892 (95% CI: 0.824–0.960). Limitations: This was a single-center study. Conclusions: For decisions regarding the treatment of IH patients, the IHReS has a higher efficiency and sensitivity than the HSS. However, the specificity of the IHReS is lower than that of the HSS.

## 1. Introduction

Infantile hemangioma (IH) is the most common benign tumor in infants and young children and its incidence can reach 4.5% [1]. Some IH sites are confined to the superficial layers of the skin and are small in size, and thus, they do not cause major damage to substantial organs and can resolve spontaneously over time without any treatment. However, based on the size or location, other IHs require treatment after being evaluated by IH specialists [2,3]. For example, large facial IHs may cause permanent scarring, disfigurement and related structural abnormalities [4,5,6,7]. IHs involving the airway may affect patients’ respiratory function, leading to life-threatening upper airway obstruction [8,9]. Vision may be affected by IHs around the eyes, resulting in astigmatism, refractive error or strabismus [10,11,12].

Propranolol was proposed as a method of treatment for IH in 2008 [13], and then a multicenter, randomized, double-blind clinical trial showed that propranolol can effectively treat IHs [14]. To date, propranolol has been rapidly and widely used. In view of the high incidence of IH and an increasing number of studies on the treatment of IH with propranolol, the standardization and rationality of IH treatment is particularly important. The IH growth curve illustrates the characteristics of its growth and development, with the fastest and most significant growth occurring between 1 and 3 months of age, and in most cases, growth reaches the maximal size at 9 months [1,15]. In 2019, the clinical practice guideline for the management of IHs formulated by the American Academy of Pediatrics (AAP) further defined some high-risk IHs that warranted concern [16] and emphasized that such high-risk IHs should be referred to IH specialists or treated in a timely manner. Therefore, for nonexpert primary physicians, it is necessary to establish a reliable scale to assess the severity of the disease and aid in decisions regarding whether the patient needs to referred to an IH specialist.

Haggstrom [17] developed the Hemangioma Severity Scale (HSS), and there have been more clinical application studies on the HSS than on any other scale. Moyakine [18] and Mull [19] conducted retrospective studies on the clinical utility of the HSS and believed that children with IHs with a total score no less than six should be referred to IH specialists. Recently, Léauté-Labrèze [20] developed the Infantile Hemangioma Referral Score (IHReS) tool for IH, which aimed to promote the more accurate and timely referral of IH patients.

These two scales provide relevant information for the referral of IH patients. However, whether patients who are referred after evaluation with one of the two scales need propranolol treatment is still unclear. The purpose of this study was to explore the impact of the evaluation results of the two scales on the therapeutic decisions regarding oral propranolol administration in IH patients. We present the following article in accordance with the STROBE reporting checklist.

## 2. Materials and Methods

### 2.1. Study Patients

This was a prospective study that included patients less than one year old who were referred by primary care physicians to our multidisciplinary vascular anomaly clinic, West China Hospital of Sichuan University, a tertiary medical institution, from May 2020 to January 2021. All patients were diagnosed with IH by our multidisciplinary vascular anomaly group (a collaboration team including IH experts in pediatric surgery, plastic surgery, pediatric dermatology and radiology) at West China Hospital of Sichuan University. The study was approved by the Ethics Committee on medical research of West China Hospital, Sichuan University and was conducted in accordance with the principles outlined in the Declaration of Helsinki. Informed consent was obtained from the patients’ parents prior to their participation in this study. The study has been registered at www.clinicaltrial.gov (accessed on 4 November 2022) (NCT03331744).

The exclusion criteria were as follows: diagnosed with other types of vascular anomalies, such as congenital hemangioma and vascular malformation; lost to follow-up during the clinical process; or withdrawal from the study for any other reason.

The authors have completed the STROBE reporting checklist. The abstract of this research was presented as an oral presentation at the 2022 International Society for the Study of Vascular Anomalies (ISSVA) Congress.

### 2.2. Data Collection

Comprehensive clinical data on the IH patients were collected, which included patient demographic information and clinical information on the sites of IHs. All the included IH patients were evaluated, and treatment decisions were made according to the definition of high-risk IH [16] formulated by the AAP and by consensus of the multidisciplinary vascular anomalies group. The patients were divided into a treatment group and a non-treatment group.

Treatments included topical timolol gel, oral propranolol and surgery. During propranolol treatment, patients were followed up at 1, 4, 12 and 24 weeks after baseline. Electrocardiogram (ECG) and blood glucose (BG) levels of the patients were closely monitored during propranolol treatment. Considering the time-sensitive nature of IHs, patients in the nontreatment group were followed up at 4, 12 and 24 weeks after baseline.

In all patients, photos and/or ultrasound images were taken of the IH lesions at baseline and during the follow-up. Three trained investigators independently assessed all the included IH patients using the HSS at every follow-up visit and took the highest score as the final value. The HSS includes both objective indicators (including size, location, risk of related structural abnormalities and complications) and subjective indicators (including pain and risk of disfigurement) of IH. As children cannot express themselves in words, the FLACC (Face, Legs, Activity, Cry, Consolability) scale was used to evaluate the degree of pain. When the patients were enrolled at the first visit, their guardians were taught the FLACC assessment immediately. Outpatients were observed by guardians at 24 h and then followed up by investigators for detailed assessment results by telephone. Inpatients were evaluated by investigators at 6, 12 and 24 h after admission. Patients with ulcers should be assessed at 1, 3, 6, 12 and 24 h after the onset of ulcers. IH patients with a total score no less than 6 were judged as needing a referral to an IH specialist [18,19]. Using the same clinical data from all the included cases, the investigators also evaluated whether the patients should be referred to the IH specialists based on the IHReS. The IHReS consists of two sections with 12 questions that evaluate referrals through visual pictures and questions. This scale is freely available through www.ihscoring.com. All the investigators received training on scale evaluations, and consistency among the observers was analyzed (Appendix A).

### 2.3. Statistical Analysis

The data were statistically analyzed by IBM SPSS Statistics 25.0 software. Although the receiver operating characteristic curve (ROC) is usually used to evaluate the accuracy of diagnosis, this study used the area under the curve (AUC) as a measure of performance to compare the efficiency of treatment decisions made using the two scales. The high-risk IHs defined by AAP were used as the gold standard to calculate the sensitivity and specificity of the two scales. The data are displayed using an ROC curve, with the X axis representing the false positive rate (FPR, or 1-specificity) and the Y axis representing the true positive rate (TPR, or sensitivity). Generally, the AUC value varied between 0.5 and 1. An AUC value > 0.9 indicates high accuracy, a value of 0.7 to 0.9 indicates medium accuracy, a value of 0.5 to 0.7 indicates low accuracy, and a value < 0.5 is regarded as an accidental result. The AUCs of the HSS and IHReS were compared by the DeLong method. When the *p* value was < 0.05, the difference was considered statistically significant.

To further illustrate the impact on treatment decision-making results, the positive predictive value (+PV), negative predictive value (−PV), positive likelihood ratio (+LR) and negative likelihood ratio (−LR) of the two scales were also calculated. The Fagan nomogram, which uses the likelihood ratio to map the pre-test probability to the post-test probability, estimates the likelihood of the final treatment decision between the two scales [21,22].

## 3. Results

### 3.1. Clinical Characteristics

In this study, a total of 266 IH patients who were referred to our hospital were included and evaluated. These 266 patients were divided into the treatment group (*n* = 215) and nontreatment group (*n* = 51). The treatment rate was 80.8%. Among them, 69 (32.1%) patients were treated with topical timolol, 9 (4.2%) patients were treated with surgical resection, and the remaining 137 (63.7%) patients were treated with oral propranolol. The clinical characteristics of the two groups are listed in Table 1. We found significant differences between the two groups in terms of age at first visit, IH lesion size, IH morphologic subtype, IH description, IH growth stage and accompanying complications. There was no statistically significant difference in sex or location of the IH. The average age of the treated patients was older than that of the patients who did not need treatment. Most patients in the treatment group were in the proliferative phase (78.6%). Sixteen (34.0%) patients with involuting IHs did not need treatment after evaluation.

### 3.2. Data Characteristics

The descriptive statistical parameters were measured after evaluation with the HSS and the IHReS (Table 2). The sensitivity of the HSS in evaluating treatment decisions after referral was 44.2%, and the specificity was 84.3%, while the sensitivity of the IHReS was 100%, and the specificity was 43.1%. The ROC curve showed that the AUC of the HSS was 0.703 (95% CI: 0.634–0.772), and the AUC of the IHReS was 0.892 (95% CI: 0.824–0.960) (Figure 1). For the patients with IH, the IHReS was more effective than the HSS in deciding whether to treat with propranolol (*p* < 0.001). The post-test probabilities corresponding to the +LR and −LR were calculated, and the Fagan nomogram was drawn (Figure 2). The post-test probabilities of −LR and +LR through the HSS were 92.3% and 75.0%, respectively, and the values of the IHReS were 88.5% and 0.0%, respectively.

## 4. Discussion

Before the development of the IHReS in 2020, there was no valid scale for evaluating IH referrals. Moreover, primary care providers may not have a comprehensive and professional understanding of IH in clinical practice. Therefore, a quantitative assessment for efficient referral is urgently needed. The utility of the HSS, the classic IH scale, has been investigated in referral studies [18,19,23]. However, the HSS is not valued or widely used by primary care providers due to its potential limitations, which require specialized knowledge of IH as well as a longer evaluation time.

Delays in referral may result in the critical time window for optimal treatment to be missed. Clinically, referrals for high-risk IH patients who are in pain, experiencing functional impairment or could suffer from permanent disfigurement or life-threatening events should be considered urgent. In this study, we used the HSS and IHReS for referral evaluations to determine the treatment for IH patients. Our results showed that the IHReS was more efficient than the HSS, with a sensitivity reaching 100%, and could refer all patients who needed treatment. Therefore, applying the IHReS for referral evaluations in clinical practice can ensure that patients who need treatment are referred to IH specialists. Early referral can provide these patients the best opportunity for obtaining timely and individualized monitoring and/or treatment.

The Fagan nomogram estimated the change in the likelihood of treatment decisions after referral evaluation. When the HSS was used, the probability for treatment increased from 80.8% to 92.3% when the results indicated that referral was needed. The probability for treatment decreased from 80.8% to 75.0% when the results indicated that no referral was required. When the IHReS was used, the probability of treatment increased from 80.8% to 88.5% when the results suggested that referral was needed. The probability of treatment was reduced from 80.8% to 0.0% when the results suggested that there was no need for referral. For patients who do not need to be referred after evaluation by the IHReS, the decision not to proceed with treatment was found to be a safe and effective choice, and there was no need for long-term monitoring by IH specialists. This is important and can avoid wasting limited medical resources. On the other hand, the IHReS can facilitate assessment by an IH specialist as soon as possible.

However, it should be noted that the IHReS was less specific than HSS, causing some IH patients who do not require treatment to be referred to hemangioma specialists. These unnecessary referrals for IH patients to receive excessive monitoring might be a burden on tertiary medical institutions.

Nonetheless, as an evaluation tool for primary care providers, the IHReS is easy to understand and operate. The use of the IHReS can greatly simplify the evaluation process and reduce the evaluation time. Although the HSS is more comprehensive, as it includes both objective and subjective evaluation indicators, it involves more professional knowledge when applied in clinical practice, and the scoring rules are more complex. In our clinic, evaluation with the IHReS took less than 1 min on average, while evaluation with the HSS took at least 2 min. Therefore, the IHReS can be more easily used and promoted by nonexpert primary physicians than the HSS.

The application of the IH assessment scale is becoming increasingly important; it can facilitate the shift from ambulatory consultation to telemedicine, not only for primary care providers referring patients but also in the context of the recent COVID-19 pandemic [24]. At the same time, scale standardization provides a strong clinical practice foundation for the development of big data/intelligent deep learning in the future.

## 5. Limitations

Our research has several limitations that should be mentioned. First, this was a single-center study conducted in a tertiary medical institution. The patients included in this study were referred from primary medical institutions. Since cases that were not reasonably referred cannot be included, selection bias may exist. Second, the morphologic subtypes and growth stages of IHs were variable. There is no fixed gold standard for treatment decisions. Treatment decisions for the patients in this study were determined according to the AAP guidelines and multidisciplinary team consultation. Finally, evaluation for referral should be conducted as early as possible when problematic IHs are detected, and some delayed referral patients, namely, those who were in the regression phase in actual clinical practice, could have also been evaluated for referral and included in our study. Although there were heterogeneous factors, our results are generalizable to clinical practice.

## 6. Conclusions

The IHReS was found to be more efficient than the HSS for decisions regarding whether to treat IH patients. The use of the IHReS for referral evaluations can comprehensively refer potential problematic IHs to specialists. Compared with the HSS, the IHReS has a higher sensitivity but lower specificity.

## Figures and Tables

**Figure 1 children-09-01851-f001:**
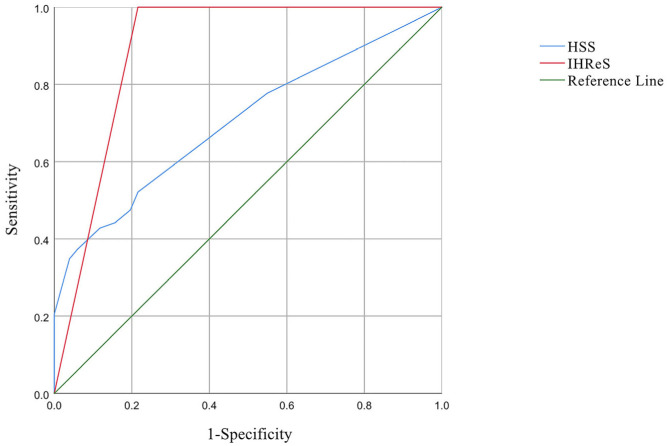
ROC comparison between the HSS and the IHReS.

**Figure 2 children-09-01851-f002:**
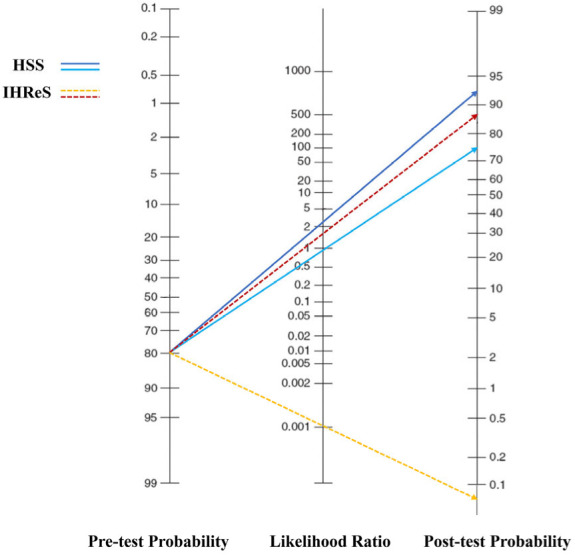
Fagan nomogram of the HSS and the IHReS. When the pre-test probability was 80.8%, the dark blue solid line indicated that the post-test probability of +LR was 92.3%, while the light blue solid line indicated that the post-test probability of −LR was 75.0% when using the HSS. The red dotted line indicated that the post-test probability of +LR was 88.5%, while the yellow dotted line indicated that the post-test probability of −LR was 0.0% when using the IHReS.

**Table 1 children-09-01851-t001:** Clinical characteristics of infantile hemangioma (IH) patients between the treatment group and the nontreatment group.

	Treatment (*n* = 215)	Nontreatment (*n* = 51)	*p* Value
Age at first visit	4.31 ± 3.19	3.76 ± 2.48	0.012
Sex, No. (%)			0.935
Male	62 (28.8%)	15 (29.4%)	
Female	153 (71.2%)	36 (70.6%)	
IH area (cm2)	12.28 ± 20.25	1.55 ± 0.87	<0.001
IH morphologic subtype, No. (%)			<0.001
Focal	161 (74.9%)	51 (100.0%)	
Segmental	51 (23.7%)	0 (0.0%)	
Multifocal	3 (1.4%)	0 (0.0%)	
IH description, No. (%)			<0.001
Superficial	154 (71.6%)	51 (100.0%)	
Mixed	53 (24.7%)	0 (0.0%)	
Deep	8 (3.7%)	0 (0.0%)	
IH growth stage, No. (%)			0.015
Proliferative	169 (78.6%)	33 (64.7%)	
Stable	15 (7.0%)	2 (3.9%)	
Involuting	31 (14.4%)	16 (31.4%)	
Location, No. (%)			0.613
Head/face	86 (40.0%)	23 (45.1%)	
Neck	13 (6.0%)	3 (5.9%)	
Limbs	57 (26.5%)	14 (27.4%)	
Trunk	49 (22.8%)	11 (21.6%)	
Perineal/perianal/genital	10 (4.7%)	0 (0.0%)	
Complication, No. (%)			
Risk of disfigurement	78 (36.3%)	5 (9.8%)	<0.001
Ulceration	23 (10.7%)	1 (2.0%)	0.050
Treatment therapy, No. (%)			
Propranolol, oral	137 (63.7%)		
Timolol, topical	69 (32.1%)		
Surgery	9 (4.2%)		

Abbreviations: m, month; IH, infantile hemangioma.

**Table 2 children-09-01851-t002:** Statistical parameters for the studied clinical measures of the HSS and IHReS.

	HSS	IHReS
Sensitivity (%)	44.2	100
Specificity (%)	84.3	43.1
+PV (%)	92.2	88.1
−PV (%)	26.4	100
+LR	2.8	1.8
−LR	0.7	0
Pre-test probability (%)	80.8	80.8
Post-test probability (%, +LR)	92.3	88.5
Post-test probability (%, −LR)	75.0	0
AUC (95%CI)	0.703 (0.634–0.772)	0.892 (0.824–0.960)

Abbreviations: HSS, the Hemangioma Severity Scale; IHReS, the Infantile Hemangioma Referral Score; +PV, positive predictive value; −PV, negative predictive value; +LR, positive likelihood ratio; −LR, negative likelihood ratio; AUC, area under the curve; CI, confidence interval.

## Data Availability

Not applicable.

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
