# Peer review of "Analysis of Therapeutic Decisions for Infantile Hemangiomas: A Prospective Study Comparing the Hemangioma Severity Scale with the Infantile Hemangioma Referral Score"

_children, 2022, doi:10.3390/children9121851_

Round 1

Reviewer 1 Report

Dear Editor,

I read with interest the paper "Analysis of therapeutic decisions for infantile hemangiomas: A prospective study comparing the Hemangioma Severity Scale with the Infantile Hemangioma Referral Score", which addresses a relatively common problem in paediatrics, both at a primary and secondary level. The aim of the work presented is well-stated, and it is clinically relevant. The paper is well-written and adequately structured.

I recommend the paper for publication without a further review from my side. 

Best regards, 

Author Response

Point 1: I read with interest the paper "Analysis of therapeutic decisions for infantile hemangiomas: A prospective study comparing the Hemangioma Severity Scale with the Infantile Hemangioma Referral Score", which addresses a relatively common problem in paediatrics, both at a primary and secondary level. The aim of the work presented is well-stated, and it is clinically relevant. The paper is well-written and adequately structured.

Response 1:  Thank you for your prompt attention to our manuscript and for your helpful comments. 

Reviewer 2 Report

This manuscript compares the Hemangioma Severity Scale and the Infantile Referral Scale (IHReS) in patients with infantile hemangioma (IH), with treatment decisions as the outcome. The superiority of the IHReS was demonstrated by appropriate statistical examination of treatment decisions as the outcome. Since the clinical sense is the outcome, it can be immediately applied to actual clinical practice. The number of cases is enough, especially the non-treatment group with 51 cases. The single center study, which is considered as a limitation, is not a problem since it could be developed into a multicenter study based on this mansucript. The heterogeneity of morphologic subtypes and growth stages is not a problem either, since it is encompassed in the clinical sense.

 As a reviewer, I cannot find anything to correct.

Author Response

Point 1: This manuscript compares the Hemangioma Severity Scale and the Infantile Referral Scale (IHReS) in patients with infantile hemangioma (IH), with treatment decisions as the outcome. The superiority of the IHReS was demonstrated by appropriate statistical examination of treatment decisions as the outcome. Since the clinical sense is the outcome, it can be immediately applied to actual clinical practice. The number of cases is enough, especially the non-treatment group with 51 cases. The single center study, which is considered as a limitation, is not a problem since it could be developed into a multicenter study based on this mansucript. The heterogeneity of morphologic subtypes and growth stages is not a problem either, since it is encompassed in the clinical sense.

Response 1:  Thank you for your prompt attention to our manuscript and thanks a lot for your helpful comments.

Reviewer 3 Report

This study is a high-volume prospective study evaluating the two scoring systems for IH.

Statistical analysis seems to be sound, and limitations are also well documented. However, I think some points need to be made clearer before it is accepted.

Line 98: How does the authors evaluate the pain of the patients under age of 1. Did they use any pain scoring system?

Line 101: I think the authors should mention about IHReS grading in more detail to deepen the reader’s understanding.

Line 168: The authors should describe about the potential limitations of HSS in more detail.

Author Response

Thank you for your prompt attention to our manuscript and for your helpful comments.
